**Data Availability Statement:** All relevant data are within the manuscript and its Supporting Information files.

# Home-based subcutaneous immunoglobulin for chronic inflammatory demyelinating polyneuropathy patients: A Swiss cost-minimization analysis

Clémence Perraudin[1]*, Aline Bourdin[1], Alex Vicino[2], Thierry Kuntzer[2], Olivier Bugnon[1,3,4], Jérôme Berger[1,3,4]

1 Community Pharmacy, Center for Primary Care and Public Health (Unisanté), University of Lausanne, Lausanne, Switzerland, 2 Nerve-Muscle Unit, Department of Clinical Neurosciences, Lausanne University Hospital (CHUV), Lausanne, Switzerland, 3 School of Pharmaceutical Sciences, University of Geneva, Geneva, Switzerland, 4 Institute of Pharmaceutical Sciences of Western Switzerland, University of Geneva, University of Lausanne, Lausanne, Switzerland

* Clemence.perraudin@unisante.ch

## Abstract

### Purpose

To compare the cost of two patient management strategies with similar efficacies for chronic inflammatory demyelinating polyneuropathy (CIDP) patients in the chronic phase: hospital-based IV immunoglobulin G (IVIg) and home-based subcutaneous immunoglobulin G (SCIg) associated with an interprofessional drug therapy management programme (initial training and follow-up).

### Methods

A 48-week model-based cost-minimization analysis from a societal perspective was performed. Resources included immunoglobulin (IVIg: 1 g/kg/3 weeks; SCIg: 0.4 g/kg/week initially and 0.2 g/kg/week in the maintenance phase), hospital charges, time of professionals, infusion material, transport and losses of productivity for patients. Costs were expressed in Swiss francs (CHF) (1 CHF = 0.93€ = US$1.10, www.xe.com, 2020/10/28).

### Results

The total costs of IVIg were higher than those of SCIg for health insurance and other payers: 114,747 CHF *versus* 86,558 CHF and 8,762 CHF *versus* 2,401 CHF, respectively. The results were sensitive to the immunoglobulin doses, as this was the main cost driver. The SCIg daily cost in the initial phase was higher for health insurance than hospital-based IVIg was, but the additional costs were compensated during the maintenance phase (from week 28). The professional costs associated with the switch were not fully covered by the insurance and were borne by the pharmacist and the nurse.

**Funding:** There was no external funding for this manuscript.

**Competing interests:** The authors have declared that no competing interests exist.

## Conclusions

SCIg for CIDP patients reinforced by an interprofessional drug therapy management programme may be a cost-effective and sustainable alternative to IVIg in the Swiss system context. From an economic perspective, this therapy alternative should be more widely supported by healthcare systems and proposed to eligible patients by professionals.

## 1. Introduction

Home-based subcutaneous immunoglobulin G (SCIg) is well established in the treatment of primary immunodeficiency diseases (PIDs) but is recent in neurology [1]. In Switzerland, only Hizentra® (CSL Behring) has been indicated since 2019 to treat chronic inflammatory demyelinating polyneuropathy (CIDP) in adults. CIDP patients generally receive hospital-based IV immunoglobulin G (IVIg), and the switch to SCIg has not yet been systematically proposed. The administration process is the same for PID and CIDP patients [2]. However, for CIDP patients, SCIg is indicated only as maintenance treatment after IVIg stabilization.

For CIDP patients, SCIg is considered to lead to similar clinical outcomes than IVIg and is well tolerated [3, 4]. SCIg is also often preferred by patients over IVIg, as it is associated to better satisfaction and quality of life [5–7]. Previous studies have shown than SCIg has the potential to be cost-effective in different countries for both PID patients [8–14] and CIDP patients [15–17]. The findings are sensitive to the national context, and more importantly, the cost of patients training and follow-up is often overlooked. Indeed, in the long-term use of SCIg, professionals stay responsible for optimal safety, effectiveness and proper medication adherence. Therefore, an interprofessional drug therapy management programme has been proposed for years by the Center for Primary Care and Public Health (Unisanté, Lausanne) to train patients with SCIg and ensure a long-term support programme for them [18, 19].

The aim of this study is to compare the cost of hospital-based IVIg and home-based SCIg associated with the patient support programme (Fig 1) to determine whether this alternative should be promoted in the Swiss context. The model and findings are transposable to other contexts adopting national unit costs.

## 2. Materials and methods

### 2.1 Study design

SCIg is indicated for CIDP patients as maintenance therapy after stabilization with IVIg. All patients started IVIg treatment at the hospital. Resources related to the stabilization phase were not estimated in this study because there is no alternative management treatment. The study assumed a standard CIDP patient in the chronic phase who was eligible for SCIg (after stabilization).

The following two management strategies were compared (Fig 1):

1. Hospital-based IVIg therapy (named "IVIg") corresponding to the Lausanne University hospital outpatient setting (CHUV, Lausanne, Switzerland),

2. Home-based SCIg therapy (named "SCIg") associated with an interprofessional drug therapy management programme during the initial phase (involving training sessions) and maintenance phase (follow-up).

Both strategies were considered to provide identical effectiveness in the treatment of CIDP in terms of relapse rates [3, 20]. We assessed the cost of the strategies over a 48-week period

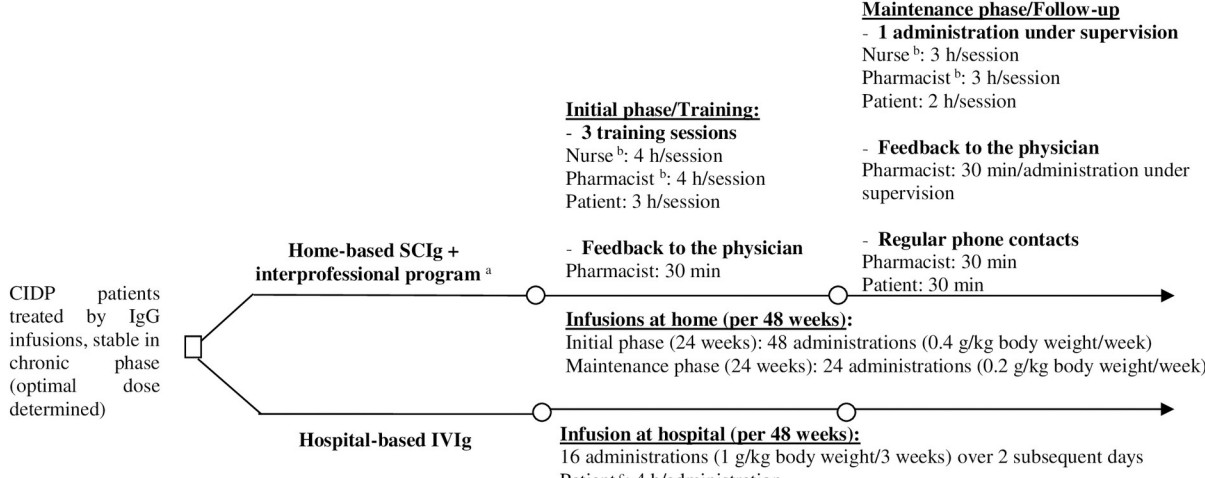

**Fig 1. Decision tree for management of CIDP patients, stable in the chronic phase, treated by IgG infusions.** [a] Interprofessional drug therapy management programme as developed and implemented at the Community Pharmacy of the Center for Primary Care and Public Health (Unisanté), University of Lausanne, Switzerland. [b] Duration included transport and time spent at patient's home. [c] Duration included transport ant time spent at hospital (infusions + waiting time + administrative time).

based on the main clinical study related to SCIg for CIDP patients [3, 4, 21] through a cost-minimization analysis. We adopted a societal perspective, i.e., we considered all costs distinguishing the payers (healthcare insurers, patients, and community). As no data from real patients were available, we adopted a simulation model whose data were mainly based on product monographs, international guidelines and expert opinions.

## 2.2 Resource use and costs

The parameters considered are shown in Table 1. The results were expressed in Swiss francs (CHF) (1 CHF = 0.93€ = US$1.10, www.xe.com, 2020/10/28).

## 2.3 Treatment

The theoretical standard patient was a 75 kg adult who was clinically stable to switch from IVIg to SCIg. The infusion doses were estimated according to product monographs [2, 22]. The standard patient received over 48 weeks:

- In IVIg therapy, 1 g/kg body weight every three weeks, i.e., 75 g in total was spread over two days in the hospital. We counted 16 infusion cycles over a 48-week period

- In SCIg therapy during the initial phase, 0.4 g/kg body weight every week, i.e., 30 g in total by week, spread over two infusions of 15 g at home. We counted 48 infusions over a 24-week period; then, during the maintenance phase, 0.2 g/kg every week, i.e., 15 g by infusion at home. We counted 24 infusion cycles over a 24-week period (Fig 1).

The choice of IgG products corresponded to Hizentra® (CSL Behring) for SCIg (the only SCIg indicated in Switzerland to treat CIDP) and Privigen® for IVIg (also manufactured by CSL Behring and the cheapest IVIg in Switzerland, allowing a conservative approach in our model). The costs of IgG corresponded to the public prices in the Swiss market [23] (see Table 1). The cost per gram from the largest package available in the Swiss market is 69 CHF for Hizentra® and 75 CHF for Privigen®.

**Table 1. Key model assumptions.**

| Parameters | Base case | Source |
|---|---|---|
| **Patient's characteristics** | | |
| Patient's weight | 75 kg | Experts' opinion |
| **Treatment—Hospital-based IVIg** | | |
| Dose IgG–Maintenance phase | 1 g/kg body weight/3 weeks | [22] |
| Number of administrations (per 48 weeks) | 16 (over 2 subsequent days) | [22] |
| Cost IgG Privigen® per package | 2,987 CHF (40 g/400 mL) | [23] |
| | 1,528 CHF (20 g/200 mL) | |
| | 784 CHF (10 g/100 mL) | |
| | 400 CHF (5 g/50 mL) | |
| Patient time[b] | 4 h/infusion | Experts' opinion |
| Hospital charges per infusion | 321 CHF/infusion | [24] |
| **Treatment—Home-based SCIg** | | [2] |
| Dose IgG–Initial phase | 0.4 g/kg body weight/week | |
| Dose IgG–Maintenance phase | 0.2 g/kg body weight/week | |
| Number of administrations | 48 per 48 weeks | [2] |
| Cost IgG Hizentra® per package | 692 CHF (10 g/50 mL) | [23] |
| | 287 CHF (4 g/20 mL) | |
| | 152 CHF (2 g/10 mL) | |
| | 84 CHF (1 g/5 mL) | |
| Dispensation fees | 3.25 per prescription form and 4.30 per medicine | [25] |
| **Materials–Home-based SCIg** | | |
| Self-infusion pump | 2,840 CHF | [26] |
| Infusion disposables | 85 CHF/infusion | |
| Other disposables (phase) | 135 CHF/24 week (initial) | Unisanté |
| | 67 CHF/24 weeks (maintenance) | |
| **Drug therapy management programme for SCIg** | | |
| **The training phase** | | |
| Nurse | 4 h[a]/session (n = 3) | Unisanté |
| Community pharmacist | 4 h[a]/session (n = 3) + 30 min feedback | |
| Patient | 3 h/session (n = 3) | |
| Fess-for-service (training by the pharmacist) | 320 CHF | [25] |
| Hourly fees for the nurse | | [27] |
| Evaluation and counselling | 80 CHF/h | |
| Examination and treatment | 65 CHF/h | |
| **The follow-up phase** | | |
| Nurse | 3 h[a]/administration under supervision (n = 1) | Unisanté |
| Pharmacist | 3 h[a]/administration under supervision (n = 1) | |
| | 1 h (feedback + regular contacts) | |
| Patient | 2 h/administration under supervision (n = 1) | |
| **Labour costs** | | |
| Nurse | 80 CHF/h | [28] |
| Pharmacist | 87 CHF/h | |
| Patient | 63 CHF/h | [29] |
| **Transport** | | |
| Distance home/hospital | 15 km | Lausanne University hospital (CHUV) |
| Travel time per way | 30 min | |

(*Continued*)

**Table 1.** (Continued)

| Parameters | Base case | Source |
| --- | --- | --- |
| Cost per km | 1 CHF | [30] |

CHF = Swiss francs, min = minutes.

[a] Duration including transport and time spent at patient's home

[b] Duration including transport and time spent at hospital (infusions + waiting time + administrative time).

The cost of hospital charges for IVIg infusions was estimated by the Swiss tariff system for medical acts [24] They covered professional time, required material and hospital overheads.

The cost of material required for home-based SCIg included fixed costs (e.g., infusion pump) and variable costs per infusion (including needles, infusion tubing, and syringes) [26]. We estimated the total cost for the amortization of the pump during the first 24 weeks in SCIg (initial phase), and we assumed that the patient did not stop SCIg during this period. In reality, if the patient stops before the end of the initial phase (i.e., before the amortization of the purchase cost of the pump), it is possible to charge the use of the pump based on the daily rental price. Both options (purchase and rental of the pump) are similar regarding their costs from the insurer perspective. Indeed, the pump currently costs 2,840 CHF and is rented at a daily price of 16.20 CHF; this means that if a patient rents a pump, it is fully paid and becomes his or her belonging after 25 weeks, approximately the duration of the 24-week initial phase [26]. The annual cost of disposables (including alcohol wipes, hand sanitizer, gauze, tape, sharps container, etc.) was estimated to be 73 CHF on average for PID patients at the Community Pharmacy of Unisanté [13]. We adjusted this cost according to the higher dosage for CIDP patients compared to PID patients (multiplying by four during the initial phase and by two during the maintenance phase). We counted dispensation fees for the validation of prescriptions by the community pharmacist for four dispensations over 48 weeks (one per trimester). It covers the pharmacists' basic cognitive services associated with the delivery of the medicines (e.g., drug information, prescription/dosage/drug-drug interactions checks or patient records) [25].

Systemic side effects of SCIg are considered similar to those of IVIg, although the frequency and severity of SCIg are generally lower [17, 20]. As each individual dose administered by SCIg is smaller than IVIg, a reduction in side effects is expected, which could reduce the total cost. From a conservative perspective and without solid cost data, we assumed the same cost of side effects management in both strategies and did not take into account this variable.

## 2.4 Drug therapy management programme to support patients with SCIg

The patient support programme costs included training sessions and follow-up delivered by the community pharmacist and the nurse (see Fig 1 and Table 1). The number of training sessions and administrations under supervision can vary based on the patients' needs. The administration process is the same for PID and CIDP patients. Hence, the standard case was estimated through data collected from a cohort of PID patients followed at the Community Pharmacy of Unisanté, who switched to home-based SCIg as part of the interprofessional programme [13]. For the delivery of this programme, the nurse's services were charged to the patient and covered by health insurance under the pricing terms and conditions [27]. The community pharmacist can charge a fee-for-service once per patient during the switch for the training (320 CHF per patient covered by the health insurance) [25]. The potential costs not covered by the insurance were estimated to be supported by the nurse and the pharmacist, and

these were not charged to the patient. As for PID patients, we did not include additional physician time or medical follow-up consultation for the SCIg compared to IVIg, as the medical care is considered the same for both of them [13].

## 2.5 Labour costs

The hourly labour costs for health professionals were derived from the salary scale for public officials in the Canton of Vaud [28] which defined salary classes (ranging from 1 to 18) and grades (ranging from 0 to 26) according to experience. This scale is applied both in Unisanté and CHUV. The annual gross salary for a mid-career clinical professional corresponded to class 11/grade 13 for a community pharmacist and class 10/grade 13 for a nurse. We assumed that nurses had the same level and grade in both strategies. A multiplier coefficient (22%) was applied to consider social security contributions and reflect the full cost to the employer. Finally, the number of annual working hours corresponded to applicable normal working hours at Unisanté and CHUV (41 hours 30 per week excluding holidays, absenteeism and down time).

## 2.6 Transport

We estimated transport costs covered by the patient commuting to the hospital for IVIg infusions and supported by the pharmacy and the nurse for training sessions requiring commuting to the patient's home. Based on our experience with PID patients at the Community Pharmacy of Unisanté and after validation by the neurologists at CHUV for CIDP patients, we assumed the standard patient's home was located 15 kilometres from the hospital, which corresponded to thirty minutes each way by car. Transport costs were estimated using an applicable compensation system in the canton of Vaud [30].

## 2.7 Indirect costs

We assumed that the standard patient was employed. The cost of productivity loss for the patient was estimated through the median gross salary per capita in Switzerland in 2016 [29] taking into account the social security contributions. The number of annual working hours corresponded to usual working hours in Switzerland (40 hours per week excluding holidays, absenteeism and down time). The total time for the patient to be administered IVIg included transport and time spent at the hospital (infusions, waiting and administrative time). In SCIg, we included the costs for time lost by the patient during both training and follow-up sessions. However, we assumed that infusions at home were self-administered outside of working hours and that the patient was free to perform various activities compatible with infusions [31].

## 2.8 Sensitivity analysis

The robustness of the results was tested by univariate sensitivity analyses, varying the parameters considered to have the greatest impact on the model output, i.e., parameters associated with the treatment [13], with other things being equal. The IgG dose for IVIg varied with the associated number of cycles, days and duration of infusions over 48 weeks, all other things being equal, from minimal dose (0.5 g/kg/6 weeks, 8 cycles over 48 weeks, 1 day infusion of 2 hours) to maximal dose (2 g/kg/3 weeks, 16 cycles over 48 weeks, 2 day infusion of 4 hours) [22]. These values were validated by expert neurologists and corresponded to real-life situations, as IgG doses for CIDP patients vary more than those for PID patients. The SCIg doses remained the same during the two treatment phases, independent of the previous IVIg doses, because these are the only ones recommended currently [2].

**Table 2. Cost estimations per patient (CHF, Swiss francs).**

| Strategy | Hospital-based IVIg | Home-based SCIg + programme [a] (altogether) | Home-based SCIg programme* (detailed by phase) | |
|---|---|---|---|---|
| Duration | 48 weeks | 48 weeks | 24 weeks (initial phase) | 24 weeks (maintenance phase) |
| Dose | 1 g/kg/3 weeks | 0.4 g/kg/week, 0.2 g/kg/week | 0.4 g/kg/week | 0.2 g/kg/week |
| **Health insurance** | **105,985** | **85,561** | **57,823** | **27,737** |
| IgG | 91,185 | 75,300 | 49,806 | 25,494 |
| Hospital charges | 14,801 | n/a | n/a | n/a |
| SCIg material | n/a | 9,133 | 7,036 | 2,098 |
| Fees for nurse | n/a | 777 | 646 | 131 |
| Fees for pharmacist | n/a | 350 | 335 | 15 |
| **Other payers** | **8,762** | **2,401** | **1,776** | **626** |
| Patient | 672 | n/a | n/a | n/a |
| Pharmacist | n/a | 1,200 | 831 | 369 |
| Nurse | n/a | 507 | 377 | 130 |
| Community | 8,090 | 695 | 569 | 126 |
| **Total costs** | **114,747** | **87,962** | **59,599** | **28,363** |
| **Daily cost covered by the health insurance** | **315** (over 336 days) | **255** (over 336 days) | **344** (over 168 days) | **165** (over 168 days) |
| **Total daily cost** | **342** (over 336 days) | **262** (over 336 days) | **355** (over 168 days) | **169** (over 168 days) |

[a] Interprofessional drug therapy management programme.

All figures are rounded up to the nearest whole number.

n/a: not applicable.

## 3. Results

Table 2 presents the costs per patient according to management strategies and phases over 48 weeks. IgG costs were the major charges, representing between 86% and 89% of costs covered by health insurance and between 79% and 87% of total costs. IVIg was the most expensive strategy (114,747 CHF *versus* 86,558 CHF over 48 weeks for IVIg and SCIg, respectively). SCIg was more expensive in the initial phase than in the maintenance phase, mainly due to the higher administered dose (0.4 g/kg/week *versus* 0.2 g/kg/week). The SCIg daily cost in the initial phase was more expensive for health insurance than IVIg, but the additional cost was compensated during the maintenance phase (from week 28). Over 48 weeks, the total savings associated with the switch to home-based SCIg for health insurance was estimated to be 21,828 CHF per patient. The professional costs associated with SCIg (training and follow-up services) were not fully covered by Swiss insurance and were borne by the pharmacist and the nurse (1,200 CHF and 507 CHF, respectively). The results were sensitive to the IgG dose used in the IVIg strategy (see Table 3).

## 4. Discussion

The study showed the potential cost-effectiveness of home-based SCIg associated with an interprofessional drug therapy management programme supporting CIDP patients from health insurance, patient and community perspectives in the Swiss healthcare context. IgG cost was the major cost driver, and SCIg was not cost-effective for CIDP patients who received IVIg with minimal IgG dose (see Table 3). However, a solution should be found in which health insurers pay for the costs of pharmacists and nurses who are not currently covered.

**Table 3. Univariate sensitivity analyses (other things being equal) (CHF, Swiss francs).**

| Strategy | Hospital-based IVIg (min IgG dose) | Hospital-based IVIg (max IgG dose) | Home-based SCIg + programme [a] (altogether, unchanged) |
|---|---|---|---|
| **Dose** | 0.5 g/kg/6 weeks | 2 g/kg/3 weeks | Initial (0.4 g/kg/week) and maintenance (0.2 g/kg/week) phases |
| **Health insurance** | **26,747** | **206,544** | **85,561** |
| IgG | 24,897 | 182,370 | 75,300 |
| Hospital charges | 1,850 | 24,175 | n/a |
| SCIg material | n/a | n/a | 9,133 |
| Fees for nurse | n/a | n/a | 777 |
| Fees for pharmacist | n/a | n/a | 350 |
| **Other payers** | **2,190** | **12,806** | **2,401** |
| Patient | 168 | 672 | n/a |
| Pharmacist | n/a | n/a | 1,200 |
| Nurse | n/a | n/a | 507 |
| Community | 2,022 | 12,134 | 695 |
| **Total costs** | **28,938** | **219,351** | **87,962** |
| **Daily cost covered by the health insurance** (over 336 days) | **80** | **615** | **255** |
| **Total daily cost** (over 336 days) | **86** | **653** | **262** |

[a] Interprofessional drug therapy management programme.

All figures are rounded up to the nearest whole number.

n/a = not applicable.

In this study, we used the static current IVIg dose/frequency regimen approved by the Food & Drug Administration and other health authorities applied in routine care in Europe for maintenance phase [22]. In practice, there is inter- and intra-individual variability depending on the patient's needs, clinical response and the course of the disease. A clinical algorithm for individualized IVIg dosing based on patient response has emerged in the literature [32, 33] but is not yet systematically implemented in routine practice. Our cost estimations based on a static model cover the majority of treatment schemes observed in the cohort study for patients on stable long-term treatment [32]. These estimations could be used in a more dynamic way to take into account possible treatment fluctuation for a patient over time and disease.

Two other cost-minimization studies compared hospital-based IVIg and home-based SCIg for CIDP patients in Italy over one year. These results confirmed the cost advantage in favour of SCIg, with the same IgG dose in both strategies, from a local health service [16] and from a societal perspective [15]. A recent study from the United States showed a cost advantage over three weeks for lower dose SCIg (0.2 g/kg/week) but not for higher dose SCIg (0.4 g/kg/week) compared to IVIg (1 g/kg/week), primarily due to the higher cost per gram of Hizentra® compared with Gamunex-C® (Grifols Therapeutics) [17]. In the Swiss context, a previous cost-minimization analysis performed for PID patients, including the same interprofessional drug therapy management programme, estimated the total savings from switching to SCIg at 9,630 CHF per patient over three years from the societal perspective. The same IgG doses were assumed for both strategies [13].

The IgG product cost represented the main total cost driver, so any difference in its cost per gram influences the advantage for either strategy (see Table 3). These costs can greatly vary from one country to another. In Switzerland, the cost per gram for all available IgG products in IV is always higher than in SC (cost per gram for the largest package [34]: Intratect® = 76

CHF, Kiovig[®] = 77 CHF, Octagam[®] = 78 CHF, Privigen[®] = 75 CHF, in IV versus Cuvitru[®] = 73 CHF, Gammanorm[®] = 67 CHF, Hizentra[®] = 69 CHF in SC) (1 CHF = 0.93€ = US$1.10, www.xe.com, 2020/10/28), which is in favour of SCIg. This is also the case in Denmark, where the difference is even larger (Privigen[®] = 111€/g *versus* Hizentra[®] = 77€/g) [35], but not in France (40€ *versus* 45€) [36] or in Belgium (44€ *versus* 48€) [37]. In the US study, the cost per gram of Gamunex-C[®] was much lower than that of Hizentra[®] (111€ *versus* 154€) [17]. These prices are negotiated at the national level and can be a political lever to encourage the switch or not. The IgG cost also depends on the doses and weight of the patient. In our analysis, we assumed weekly IgG doses for SCIg recommended for a standard patient [3].

The choice of a cost-minimization analysis was consistent with the state of current knowledge. As subcutaneous alternatives are new for CIDP treatment, we lack clinical and follow-up data for CIDP patients switching to home-based SCIg. The phase 3 randomized controlled trial PATH compared low-dose (0.2 g/kg/week) and high-dose (0.4 g/kg/week) SCIg to placebo with CIDP patients responding to IVIg [3]. No data are available on the direct comparisons between IVIg and SCIg. Moreover, the real-life data on the probability of stopping the SCIg alternative are not available. Patients may stop due to relapse or local side effects. In the PATH trial, the discontinuation rate of SCIg during the 24-week period was defined as the proportion of patients who relapsed or were removed from the study for any other reason [3]. The discontinuation rate was 63% (n = 36/57) in the placebo group, 39% (n = 22/57) in the low-dose group (0.2 g/kg/week) and 33% (n = 19/58) in the high-dose group (0.4 g/kg/week) (p = 0.0007). Of all the patients, 6 withdrew consent because of issues with SCIg infusions (mild local reactions, did not feel comfortable with the SCIg technique, no longer wanted to participate because of a need to travel abroad). The support of healthcare professionals in the training and follow-up of patient switching (e.g., the drug therapy management programme) could reduce these withdrawals. In the case of relapse of a stable patient (0.2 g/kg/week), it is recommended to return to a higher subcutaneous dose of 0.4 g/kg/week [4]. If the patient does not respond or if a relapse occurs at this dose, it is recommended to return to IVIg with the initial dose before the switch to SCIg. In this case, the SCIg alternative is no longer recommended. From an economic perspective, if the patient discontinues the SCIg before the 28 weeks after the switch, the setting up costs will be lost for the health insurance and liberal professionals, with potential additional management costs of the withdrawal (e.g., relapse, medical visits). In this case, the rental of the infusion pump in SCIg until this purchase is an ideal option.

For many patients, the home setting is the preferable therapy option leading to more autonomy and flexibility in SCIg [38, 39] or IVIg [40], as long as training and follow-up by professionals are provided [41]. Moreover, in a pandemic situation, the home setting avoids bringing chronic patients to the hospital for treatment. Most authors recommend presenting each patient with the potential advantages and disadvantages of both alternatives with respect to individual preferences and competencies [17, 19].

Our study did not investigate the cost-effectiveness of home-based IVIg, which is uncommon in Switzerland but appears to be an acceptable efficient alternative in other countries such as in France for patients with autoimmune neuropathies [40]. We considered that this management strategy is more expensive than SCIg for the health insurers accounting avoided hospital costs but a higher IgG cost and long-term costs associated to recurrent home-based care. We assumed it as less advantageous for patients too, because they cannot carry out daily activities during IV treatment compared to SCIg (i.e. same loss of productivity than in the hospital, except for commuting to the hospital).

Some limitations need to be acknowledged. The simulation model estimating the results is based on assumptions, and conditional parameter values. Future researches should consider

experimental prospective studies with real data collection (such as dosage, efficacy, quality of life, cost data). We adopted as much as possible a conservative approach related to the economic aspects of SCIg to avoid conclusions that were too favourable.

## 5. Conclusions

The results of the simulation model support the promotion of the switch to home-based SCIg in the Swiss system context with an example of a patient support program involving several healthcare professionals. The final management strategy decision should be discussed between the patient and the clinicians considering these economic aspects, as well as pharmacokinetics, administration procedures, adverse events and patient characteristics and preferences. SCIg should be more widely supported by healthcare systems and proposed to eligible patients by professionals.

## Supporting information

**S1 Data.**
(XLSX)

## Author Contributions

**Conceptualization:** Clémence Perraudin, Olivier Bugnon, Jérôme Berger.

**Data curation:** Clémence Perraudin, Aline Bourdin, Alex Vicino, Thierry Kuntzer, Jérôme Berger.

**Formal analysis:** Clémence Perraudin, Olivier Bugnon.

**Investigation:** Clémence Perraudin.

**Methodology:** Clémence Perraudin, Aline Bourdin, Alex Vicino, Thierry Kuntzer, Olivier Bugnon, Jérôme Berger.

**Supervision:** Olivier Bugnon, Jérôme Berger.

**Validation:** Clémence Perraudin, Thierry Kuntzer.

**Writing – original draft:** Clémence Perraudin, Olivier Bugnon, Jérôme Berger.

**Writing – review & editing:** Aline Bourdin, Alex Vicino, Thierry Kuntzer.

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
