## [Decision Letter · Decision Letter 0]

23 Sep 2020

PONE-D-20-25250

Home-based subcutaneous immunoglobulin for chronic inflammatory demyelinating polyneuropathy patients: a Swiss cost-minimization analysis

PLOS ONE

Dear Dr. Perraudin,

Thank you for submitting your manuscript to PLOS ONE. After careful consideration, we feel that it has merit but does not fully meet PLOS ONE’s publication criteria as it currently stands. Therefore, we invite you to submit a revised version of the manuscript that addresses the points raised during the review process.

We look forward to receiving your revised manuscript.

Kind regards,

Marcello Moccia

Academic Editor

PLOS ONE

Journal Requirements:

2.Thank you for stating the following financial disclosure:

 No. The funders had no role in study design, data collection and analysis, decision to publish, or preparation of the manuscript.

Additional Editor Comments (if provided):

I would recommend authors carefully address the comments on dosing and frequency, also considering new sub-analyses.

Reviewers' comments:

Reviewer's Responses to Questions

**Comments to the Author**

1. Is the manuscript technically sound, and do the data support the conclusions?

Reviewer #1: Partly

Reviewer #2: Yes

2. Has the statistical analysis been performed appropriately and rigorously? 

Reviewer #1: Yes

Reviewer #2: Yes

3. Have the authors made all data underlying the findings in their manuscript fully available?

Reviewer #1: Yes

Reviewer #2: Yes

4. Is the manuscript presented in an intelligible fashion and written in standard English?

Reviewer #1: Yes

Reviewer #2: Yes

5. Review Comments to the Author

Reviewer #1: This is description of a simulation model of SCIg vs IVIg for CIDP in the Swiss healthcare system.

The use of a simulation model poses the problem of being exclusively theoretical not taking into account 2 important issues which are variability of dose and frequency needs as well as individual variations in those needs in the same subject.

The authors quote a selected reference list and fail to take into account that IVIg dose in practice does not need to be at 1g/kg/3w in many/most patients- they do not consider dosing algorithms proposed in recent literature, nor do they tackle or even mention the issue of using actual weight vs. corrected weight through use of ideal body weight.

The doses of SCIg used are from the 2018 study whereas the IVIg dose from a 2008 RCT, and if the latter fail to be considered in the setting of this literature using dosing algorithms, comparison will not be meaningful.

The clinical response cannot compare the assumption that 1 g per kg of IVIg/3w after loading, will necessarily be equivalent to 0.4, then 0.2g/kg/w of SCIg.

Therefore, we are not comparing what is comparable, and with just the dosing, the study outcome becomes obvious.

This put in the context of a purely hypothetical setting, brings little added knowledge or value, to this important question.

Reviewer #2: The manuscript is a cost-minimization study comparing hospital-based IVIg and home-based sub-cutaneous immunoglobulin (SCIg) for patients with Chronic Inflammatory Demyelinating Polyneuropathy (CIDP) in Switzerland. The Authors demonstrated that SCIg for CIDP patients reinforced by an interprofessional drug therapy management programme is a cost-effective and sustainable alternative to IVIg in the Swiss system context. Furthermore, the authors propose that home-based sub-cutaneous immunoglobulin (SCIg) should be more widely supported by healthcare systems and proposed to eligible patients by professionals. The results are convincing and could reduce the cost of the treatment of CIDP, a significant health issue

Minor points:

1)    The Authors should provide more evidence and details on why they select Privigen for IVIg. Given that IVIg can be manufactured by many other companies and they difference cost, the selection of the manufacturer can influence the result and the conclusion that the Authors are making. Hence it is important to justify their choice.

2)    The Authors should also consider or at least discuss the cost of IVIg infusion at home. This is also an alternative and in this case, the treatment cost should not include the cost for the hospital charges for infusion.

3)    The Authors assumed that infusion at home we self-administered outside the working hours also in Switzerland however they reference a study (ref 31) done in the United States. They should justify and provide evidence that this assumption is indeed the case for patients receiving subcutaneous immunoglobulin in Switzerland.

4)    The Authors should stress the advantages of the home-based subcutaneous immunoglobulin in the current COVID emergency.

6. PLOS authors have the option to publish the peer review history of their article (what does this mean?). If published, this will include your full peer review and any attached files.

Reviewer #1: No

Reviewer #2: No

---

## [Author Response · Author response to Decision Letter 0]

28 Oct 2020

Reviewer #1: This is description of a simulation model of SCIg vs IVIg for CIDP in the Swiss healthcare system.

The use of a simulation model poses the problem of being exclusively theoretical not taking into account 2 important issues which are variability of dose and frequency needs as well as individual variations in those needs in the same subject.

The authors quote a selected reference list and fail to take into account that IVIg dose in practice does not need to be at 1g/kg/3w in many/most patients- they do not consider dosing algorithms proposed in recent literature, nor do they tackle or even mention the issue of using actual weight vs. corrected weight through use of ideal body weight.

The doses of SCIg used are from the 2018 study whereas the IVIg dose from a 2008 RCT, and if the latter fail to be considered in the setting of this literature using dosing algorithms, comparison will not be meaningful.

The clinical response cannot compare the assumption that 1 g per kg of IVIg/3w after loading, will necessarily be equivalent to 0.4, then 0.2g/kg/w of SCIg.

Therefore, we are not comparing what is comparable, and with just the dosing, the study outcome becomes obvious.

This put in the context of a purely hypothetical setting, brings little added knowledge or value, to this important question.

Response : Thanks a lot for this interesting consideration. We choose a theoretical model to simulate the cost linked to the switch from IVIg to SCIg for stable patients using the referent dose/frequency scheme approved by the FDA (and other health authorities) and used in Switzerland for CIDP patients (according to the professional information of each product). 

We agree that in practice, a wide variety of dosing regimens have been reported between patients and sometimes for the same patient according to the course of the disease and the clinical response. We assessed the impact of these variabilities in our sensitivity analyses to cover the majority of the clinical situations, but that’s true with static regimens. 

The clinical algorithm of the recent literature to optimize IVIg dose based on patient response (Lunn et al. Journal of the Peripheral Nervous System 2016;21:33–37) is not yet systematically implemented in practice. However, this approach is already considered in our model: in Lunn et al., the mean observed dose in maintenance phase for 63 patients (CIDP and MMN with no significant difference) was estimated to 1.4g/kg/ 4.3 weeks (= 0.325g/kg/week). Our cost estimations based on the static model cover the majority of dose/frequency situations for these patients [main analysis: 1g/kg/3weeks = 0.333/kg/week; minimum scenario: 0.5g/kg/6 weeks = 0.083g/kg/week; maximal scenario : 2g/kg/3 weeks = 0.667g/kg/week]. We thank you for this comment and add this interesting reflection in the discussion section (lines 217-224)

Considering the second point (actual weight vs. corrected weight), a multiplier coefficient of 1.37 is applied during the switch from IVIg to SCIg for patients with primary immunodeficiency in the United-States (for patient switching from Privigen®, as proposed in our simulation). However, no multiplier coefficient is requested for CIDP patients (according to the professional information of Hizentra®). Consequently, no coefficient was applied for corrected weight in our simulation model. 

Reviewer #2: The manuscript is a cost-minimization study comparing hospital-based IVIg and home-based sub-cutaneous immunoglobulin (SCIg) for patients with Chronic Inflammatory Demyelinating Polyneuropathy (CIDP) in Switzerland. The Authors demonstrated that SCIg for CIDP patients reinforced by an interprofessional drug therapy management programme is a cost-effective and sustainable alternative to IVIg in the Swiss system context. Furthermore, the authors propose that home-based sub-cutaneous immunoglobulin (SCIg) should be more widely supported by healthcare systems and proposed to eligible patients by professionals. The results are convincing and could reduce the cost of the treatment of CIDP, a significant health issue

Minor points:

1) The Authors should provide more evidence and details on why they select Privigen for IVIg. Given that IVIg can be manufactured by many other companies and they difference cost, the selection of the manufacturer can influence the result and the conclusion that the Authors are making. Hence it is important to justify their choice.

Response : Thank you for this comment. Hizentra® and Privigen® are both product manufactured by CSL Behring. Hizentra® was the first SCIg indicated in Switzerland for CIDP patients and this choice allowed a comparison between two products from the same manufacturer. We added this comment in the method section (line 109). Privigen® is also the cheapest IVIg on the Swiss market (see the comparison of gram cost of different IVIg products in the discussion section, lines 234-244) and therefore allowed a conservative approach in our model. 

2) The Authors should also consider or at least discuss the cost of IVIg infusion at home. This is also an alternative and in this case, the treatment cost should not include the cost for the hospital charges for infusion.

Response : IVIg at home is not common in Switzerland but is indeed an efficient strategy implemented in other countries. We were already discussing the cost-benefit for this strategy in terms of the cost of care compared to other ones in the discussion section (lines 270-276). Considering the costs of IVIg and of professionals coming at home for patients’ care, this option is economically not more interested than home-based SCIg.

3) The Authors assumed that infusion at home we self-administered outside the working hours also in Switzerland however they reference a study (ref 31) done in the United States. They should justify and provide evidence that this assumption is indeed the case for patients receiving subcutaneous immunoglobulin in Switzerland.

Response : Thank you for this interesting comment. We did not find quantitative and robust data in Switzerland regarding this point. Nevertheless, we observed in our cohort of patients with primary immunodeficiency that no patient had to reduce their working time because of home-based SCIg infusions (we observed that most of the patients perform SCIg infusion in the evening or during the week-end). We therefore assume that this hypothesis is also valid in Switzerland. It would nevertheless be interesting to verify this hypothesis in a cohort of CIDP patients.

4) The Authors should stress the advantages of the home-based subcutaneous immunoglobulin in the current COVID emergency.

Response : We add this argument in the discussion section (line 267): “Moreover, in a pandemic situation, the home setting avoids bringing chronic patients to the hospital for treatment”.

---

## [Decision Letter · Decision Letter 1]

6 Nov 2020

Home-based subcutaneous immunoglobulin for chronic inflammatory demyelinating polyneuropathy patients: a Swiss cost-minimization analysis

PONE-D-20-25250R1

Dear Dr. Perraudin,

We’re pleased to inform you that your manuscript has been judged scientifically suitable for publication and will be formally accepted for publication once it meets all outstanding technical requirements.

Kind regards,

Marcello Moccia

Academic Editor

PLOS ONE

Additional Editor Comments (optional):

Reviewers' comments:

Reviewer's Responses to Questions

**Comments to the Author**

1. If the authors have adequately addressed your comments raised in a previous round of review and you feel that this manuscript is now acceptable for publication, you may indicate that here to bypass the “Comments to the Author” section, enter your conflict of interest statement in the “Confidential to Editor” section, and submit your "Accept" recommendation.

Reviewer #2: All comments have been addressed

2. Is the manuscript technically sound, and do the data support the conclusions?

Reviewer #2: Yes

3. Has the statistical analysis been performed appropriately and rigorously? 

Reviewer #2: Yes

4. Have the authors made all data underlying the findings in their manuscript fully available?

Reviewer #2: Yes

5. Is the manuscript presented in an intelligible fashion and written in standard English?

Reviewer #2: Yes

6. Review Comments to the Author

Reviewer #2: The Authors have adequately addressed all comments raised.

Major and Minor points were addressed by the Authors

7. PLOS authors have the option to publish the peer review history of their article (what does this mean?). If published, this will include your full peer review and any attached files.

Reviewer #2: No

---

## [Editor Report · Acceptance letter]

13 Nov 2020

PONE-D-20-25250R1 

Home-based subcutaneous immunoglobulin for chronic inflammatory demyelinating polyneuropathy patients: a Swiss cost-minimization analysis 

Dear Dr. Perraudin:

I'm pleased to inform you that your manuscript has been deemed suitable for publication in PLOS ONE. Congratulations! Your manuscript is now with our production department. 

Kind regards, 

on behalf of

Dr. Marcello Moccia 

Academic Editor

PLOS ONE